# Movement Behavior and Health Outcomes among Sedentary Adults: A Cross-Sectional Study

**DOI:** 10.3390/ijerph20054668

**Published:** 2023-03-06

**Authors:** Federico Arippa, Athena Nguyen, Massimiliano Pau, Carisa Harris-Adamson

**Affiliations:** 1School of Public Health, University of California, Berkeley, CA 94720, USA; 2Department of Mechanical, Chemical and Materials Engineering, University of Cagliari, 09123 Cagliari, Italy; 3Department of Medicine, University of California, San Francisco, CA 94143, USA

**Keywords:** office workers, movement behavior, sedentary workers, musculoskeletal discomfort, cardiometabolic health indicators

## Abstract

Background: Sedentary behavior, which is highly prevalent among office workers, is associated with multiple health disorders, including those of the musculoskeletal and cardiometabolic systems. Although prior studies looked at postures or physical activity during work or leisure time, few analyzed both posture and movement throughout the entire day. Objective: This cross-sectional pilot study examined the movement behavior of sedentary office workers during both work and leisure time to explore its association with musculoskeletal discomfort (MSD) and cardiometabolic health indicators. Methods: Twenty-six participants completed a survey and wore a thigh-based inertial measuring unit (IMU) to quantify the time spent in different postures, the number of transitions between postures, and the step count during work and leisure time. A heart rate monitor and ambulatory blood pressure cuff were worn to quantify cardiometabolic measures. The associations between movement behavior, MSD, and cardiometabolic health indicators were evaluated. Results: The number of transitions differed significantly between those with and without MSD. Correlations were found between MSD, time spent sitting, and posture transitions. Posture transitions had negative correlations with body mass index and heart rate. Conclusions: Although no single behavior was highly correlated with health outcomes, these correlations suggest that a combination of increasing standing time, walking time, and the number of transitions between postures during both work and leisure time was associated with positive musculoskeletal and cardiometabolic health indicators among sedentary office workers and should be considered in future research.

## 1. Introduction

Sedentary time has been steadily increasing while physical activity has been decreasing worldwide [1]. The World Health Organization (WHO) reported that 31% of people who are 15 years or older take part in less than 2.5 h per week of moderate activity and roughly 3.2 million deaths per year are associated with such sedentary lifestyles [2]. The impact of sedentary behavior is vast and includes higher healthcare costs, loss in productivity, and increased disability-adjusted life-years (DALYs) [3]. Further, sedentary behavior continues to grow as leisure activities include computers or screen watching [4,5,6,7] and the number of sedentary jobs increases at the expense of physically active ones [8]. With an average of 8 to 12 h per day [9,10], office employees are among those workers that are most sedentary, accounting for almost 81.8% of their work hours.

Prior literature indicates that sedentary time is highly associated with musculoskeletal discomfort (MSD) [11,12,13,14,15] and adverse cardiometabolic outcomes [16,17,18,19,20,21]. Prolonged sitting time has a negative impact on resting heart rate, adiposity, vascular function [17,20], plasma glucose, HDL-cholesterol, and triacylglycerol [19]. In addition to cardiometabolic outcomes, prolonged sitting was associated with diminished endothelial function in the leg vasculature and more frequent urinary tract symptoms [22,23,24,25]. Prolonged sitting also leads to increased and sustained intradiscal pressure since compressive forces are higher when sitting compared with standing [13,15,26], which is detrimental to the hydration and nutrition of the intervertebral disc [14], thus having a negative impact on the low back both in terms of musculoskeletal pain and biomechanical load when sustained for long durations [11,27]. However, periodic breaks in sedentary time that include brief episodes of standing or walking, as well as higher total physical activity, were found to be helpful to nourish the nucleus pulposus and intervertebral disc [26] and to reduce adiposity [19].

Studies documented the inverse relationship between increased standing time and all-cause mortality rates among people who primarily sit [28,29]; even 10 breaks per day were associated with positive cardiometabolic outcomes, including decreased waist circumference, systolic blood pressure, triglycerides, glucose, and insulin and with increased HDL-cholesterol [16]. Thus, experts recommended that office-based workers should include at least 2 h per day of standing and walking during working hours [30]. Nonetheless, it is important to distinguish between periodic and prolonged standing, the latter of which has harmful effects as well [31,32,33,34,35,36,37,38]. Since numerous studies have associated prolonged standing with poor cardiometabolic and musculoskeletal outcomes [34], the recommendation to increase standing time is unique to those who spend most of their time in a seated position, and even then, frequent short intervals of standing time are recommended throughout the day versus one long bout.

Increasing movement through posture transitions, walking, or even micromovements while seated, which is defined by small changes in one’s position without large changes in postures, was suggested as a potential way to mitigate the negative health effects on the musculoskeletal and cardiometabolic systems associated with prolonged sedentary behavior [39,40,41,42,43,44]. WHO guidelines recommend that adults should undertake 150 to 300 min of moderate-intensity physical activity, 75–150 min of vigorous-intensity physical activity, or some equivalent combination of moderate-intensity and vigorous-intensity aerobic physical activity per week [45]. Walking was identified as the most effective method to improve MSD and cardiometabolic outcomes among sedentary workers [46,47,48] since walking changes the demands on the musculoskeletal system and increases energy expenditure compared with both sitting and standing [49]. Previous research shows that more frequent walking during a sedentary time was negatively associated with body mass index (BMI), waist circumference, 2 h plasma glucose, triglycerides [50,51,52,53], and MSD [54]. However, the frequency and duration required for maximum benefit remain unclear. In addition, transitioning between postures may be beneficial to both musculoskeletal and cardiometabolic health. Sit–stand workstations were implemented as a way to support transitions between sitting and standing [11], and task-based walking, such as “walk and talk” meetings, was implemented as a way to transition between sitting or standing and walking. Micromovements can be quantitatively assessed, for instance, by tracking the trunk’s center of pressure using force platforms or pressure-sensitive mats [42,43,55,56,57]. Although micromovements were suggested as a coping strategy to reduce discomfort once it has developed [42,43,57,58], people who perform a higher number of micromovements proactively have a lower probability of developing low back pain; thus, interventions designed to increase micromovements before pain develops may prevent or prolong MSD from developing [59,60,61,62,63].

### A Comprehensive Movement Behavior Model

Considering prior studies on movement, MSD, and cardiometabolic outcomes [11,12,13,14,15], we present a framework (Figure 1) to assess daily movement behavior among sedentary workers to determine which measurements are most important to optimize their health outcomes. The first strategy was to determine the total time spent sitting, standing, and walking each day. This was based on the evidence that prolonged sitting time is associated with MSD symptoms [13,15,26], and that bouts of standing and walking of suitable duration can mitigate discomfort and improve cardiometabolic measures [26,30,48,52,53,64,65,66,67]. These variables are interrelated; as one sits less, they will stand or walk more.

The second strategy was to describe the movement that occurs while sitting, standing, or walking. While sitting and standing, micromovements can be quantified by sway patterns (sway path and area), mean pressure, and in-posture movements that capture quick shifts of the body [42]. Walking, which is a general term that can differ substantially by person or environment, can also be further defined by quantifying step count and cadence, which captures the rate of steps.

A third strategy to quantify movement behavior was to measure the number of transitions between sitting, standing, and walking, which provides feedback on the pattern of whole-body posture changes. Transitions are defined by moving from sitting to standing, standing to walking, or sitting to walking (Figure 1). As noted above, numerous studies identified benefits associated with transitions and more work is being done to understand how to optimize their frequency and timing [13,51,59,68,69]. Although transitions are often prompted by discomfort, this reactive approach may be “too little too late” to mitigate MSD. Ideally, transitions should occur proactively as part of a comprehensive strategy to prevent MSD and contribute to one’s overall daily energy expenditure.

Although prior research investigated the relationships between certain measures of movement behavior, musculoskeletal discomfort, and adverse cardiometabolic outcomes [11,12,13,14,15,26,30,48,52,53,64,65,66,67], it is still unclear which, singularly or in combination, are the most impactful. Additionally, little is known about whether different movement strategies are more important during work or leisure time.

In summary, although multiple strategies were identified as ways to mitigate the negative health effects associated with prolonged sitting among sedentary office workers, a comprehensive approach to quantifying these movements is needed to understand which combinations and patterns of movement are most important for optimizing musculoskeletal and cardiometabolic health. By measuring movement behavior more comprehensively and consistently during both work and leisure time, we can help sedentary office workers to optimize their movement behavior throughout their day. Furthermore, consistent measurement of movement behaviors may increase our understanding of the effectiveness and efficacy of interventions designed to increase movement in sedentary office workers. Therefore, in addition to developing the comprehensive Movement Behavior Model used to quantify sedentary postures and movement, the aim of this pilot study was twofold: (1) investigate the existence of differences in movement behavior metrics between those with MSD and those without MSD; and (2) examine the relationships between movement behavior metrics (during work and leisure time), MSD, and cardiometabolic outcomes.

## 2. Methods

### 2.1. Participants

In total, 31 office workers at the University of California, Berkeley applied to take part in this cross-sectional pilot study. Among them, twenty-six met the inclusion criteria and were enrolled in this study. Recruitment methods included posting flyers throughout campus and sending emails through department listservs. The inclusion criteria specified that participants must have a sit–stand desk, work at the desk for at least thirty hours per week, and be capable of standing for at least twenty minutes. Exclusion criteria included any MSD or illness that would prevent the worker from standing while working at their desk. This study was approved by the University of California, Berkeley Committee for Protection of Human Subjects (protocol code 2019-10-12607).

### 2.2. Procedure

All data were collected at the UC Berkeley campus in participants’ offices at the beginning of their work shifts. Upon arrival, participants signed an informed consent form and anthropometric measurements were collected. Participants were informed that they could withdraw at any time during the study without consequences. Subjects were asked to complete a baseline survey through a Qualtrics link sent via SMS text message. The survey gathered data on demographic characteristics, physical activity, and MSD using the 0 to 10 Numeric Pain Rating Scale (NRS).

Additionally, participants were instructed to wear the activPAL monitor (Glasgow, UK) on their thighs for at least 48 h to record activity and posture data [70], and wore an Actiheart heart rate monitor (Boerne, TX, USA) and Spacelabs blood pressure cuff (Snoqualmie, WA, USA) for 24 h.

### 2.3. Measures

Height and weight were measured by means of an ultrasonic digital height meter (Soehnle 5003, Soehnle, Germany) and a digital scale (RE310, Wunder, Italy), respectively. BMI was calculated by dividing the individual’s body mass (expressed in kilograms) by their stature (meters squared); values of BMI lower than 18.5 identify underweight, between 18.5 and 25 is considered in the healthy range, while 25 to 30 and higher than 30 fall in the overweight and obese ranges, respectively. Regarding the hip–waist measurement, the WHO protocol was followed [71]; a lower hip–waist ratio indicates a healthier distribution of body fat and lower risks of health problems (0.95 or less for men and 0.80 or less for women [72]).

Activity and posture were quantified using step count and the duration of time spent sitting, standing, and walking. Postural transitions were defined as changes between sitting, standing, and walking. Activity and posture data were sampled at a rate of 20 Hz.

The MSD scores were grouped into four regions: (1) head, neck, and shoulders; (2) upper and lower back; (3) hips, knees, feet, and ankles; and (4) elbows, hands, and wrists. A composite MSD score was generated by summing the NRS scores across the four regions for a maximum score of 40.

Cardiometabolic data, including heart rate (HR), mean arterial pressure (MAP), and pulse pressure (PP) was based on 1 min and 30 min (6 a.m.–10 p.m.) or 60 min (10 p.m.–6 a.m.) sampling rates, respectively. Resting HR was calculated by taking the average of the five lowest values throughout the day, while the average HR was calculated using the data from the 24 h period; a lower HR (optimal range 60–80 bpm [73]) is an indicator of cardiovascular health. The MAP, which is the average blood pressure per cardiac cycle, was calculated by doubling the diastole measurement, adding the systole measurement, and dividing the value by three [74]; similarly to HR, a low MAP identifies healthier subjects. The PP represents the difference between systolic blood pressure and diastolic blood pressure [75]; with regular physical activity, the elasticity of blood vessels improves, resulting in a lower PP, both during rest and exercise.

### 2.4. Analysis

According to the self-reported questionnaire on MSD, people with a composite pain score of 2 or greater were categorized in the group titled “MSD”, whereas those who had a composite MSD score less than 2 in all four regions were sorted into the “NO-MSD” group. Self-reported work hours from the baseline survey were also used to stratify the movement behavior by work and leisure time for each participant.

Two-tailed independent-sample *t*-tests were used to assess differences in demographics, cardiometabolic outcomes, and movement activity (daily, work, and leisure) between the MSD and NO-MSD groups.

Spearman correlation coefficients were used to understand whether activity levels at work matched those during leisure time, and to explore the relationship between motor behavior, MSD, and cardiometabolic data. For absolute values of r, 0.0 to 0.39 was considered weak, 0.4 to 0.69 was moderate, 0.7 to 0.99 was strong, and 1 was perfect [76].

All analyses were completed using SPSS v26. An alpha of 0.05 was used as the threshold for significance and all independent variables were between subjects.

## 3. Results

Demographic analysis (Table 1) showed that most of the participants were female, and the average age of the subjects was 33.2 ± 9.4 years old. Only two participants had a current medical condition and one subject had diabetes. One participant had a previous injury.

Scores for MSD reported during the baseline survey are reported in Table 2. Daily, work, and leisure activity data of participants are presented in Table 3. There were minimal differences between the MSD and NO-MSD activity levels; however, participants in the MSD group transitioned significantly more frequently compared with those in the NO-MSD group during working hours.

The activity data (Table 3) and composite MSD score (Table 2) were used for the analyses in Table 4. Moderate positive correlations were found between the composite MSD score and transitions, while a negative relationship was evidenced with time spent standing (Table 4).

Correlations between movement behavior metrics during work and leisure are presented in Table 5. Time spent in a seated position during leisure was negatively associated with the number of steps and the standing and walking time at work. Sleeping time was positively correlated with the number of steps, transitions, and walking time at work.

The average cardiometabolic measurements showed moderate correlations with activity measures (Table 6). In particular, transitions were negatively correlated with BMI and heart rate; walking time during leisure was positively associated with average pulse pressure.

## 4. Discussion

In this cross-sectional pilot study, we looked at the relationships between movement behavior measures, pain, and cardiometabolic outcomes to inform future studies on which measurements may be important indicators of health during work and leisure time.

Overall, the results showed that participants were primarily sedentary, sitting for an average of more than 9 h per day. People with MSD transitioned significantly more throughout their working time relative to people without MSD. This is consistent with previous publications stating that people suffering from back pain tended to move more frequently than those without pain, as frequent posture changes provided relief and rest for passive and active structures that accumulate pressure during static postures, especially of the spine [11,26]. However, despite the participants in the MSD group transitioning more frequently, they also spent one hour less standing compared with those in the NO-MSD group, though this difference was not statistically significant. It is unclear whether the reduced amount of standing contributed to increased MSD, or whether perceived MSD contributed to less standing. The average amount of standing time per hour at work was 18 min in the MSD group and 24 min in the NO-MSD group. Therefore, on average, both groups stood more than what was reported in other studies, and what was recommended in previous papers [77,78]. Since the development of pain, especially low back pain, can occur during relatively short standing bouts [79,80,81,82,83,84,85,86], it is possible that participants with MSD spent less time in a standing posture because the development of pain was quicker than those without MSD [87]. This trend was also confirmed when investigating the correlation between MSD score and movement behavior indicators since MSD was positively correlated with postural transitions at work and negatively correlated with daily standing time. It is possible that the recommendation for people with MSD should focus on more posture transitions to avoid static standing postures that can result in increased pain and discomfort levels.

Regarding the relationship between movement behavior and cardiometabolic outcomes, we found negative associations for leisure and daily (leisure and work) transitions with BMI, resting heart rate, and average heart rate. This may indicate that in addition to reducing MSD, changing posture more frequently throughout the day (particularly during leisure time) may be important for cardiometabolic health outcomes. Our findings are consistent with prior studies that examined the relationships between breaks that interrupt sedentary time and improve cardiometabolic outcomes [50,51,88] emphasizing the importance of "sitting less and moving more". Although the time spent standing and walking did not have any relationship with cardiometabolic outcomes in our population, it is possible that our analysis was underpowered and included a generally healthy population that stood and walked more than other study populations [30,47,48,52,53,64]. Overall, in addition to time spent sitting, standing, and walking [89], our findings indicate that quantifying daily leisure and work transition metrics might be beneficial to optimizing MSD and cardiometabolic health. While further investigation is needed to ascertain the relative importance of these variables, these data and the developed Movement Behavior Model may serve as a general model for quantifying and further clarifying activity levels, MSD, and cardiometabolic risk in office workers.

In this study, it was also evaluated whether participants’ movement behavior during work was consistent with their movement behavior during leisure time since movement behavior during both work and leisure time was proposed as an important strategy to reduce the risk of cardiometabolic disorders and MSD [90,91]. The results showed that participants who were more sedentary at work were also more sedentary during leisure and had less sleep duration than those who were active. Although this is consistent with prior studies that found moderate physical activity to contribute toward improving both sleep quality and duration, the relationship here was not bound to moderate physical activity and included all non-sitting time [92]. If, in fact, more non-sitting time and transitions throughout the day improve sleep, this could be an additional benefit to encourage more movement during work and leisure time, especially in a sedentary population. Further studies should be performed to understand how interventions that reduce sedentary behavior at work also impact sedentary behavior during leisure time, and whether the increase in movement impacts sleep.

It should be noted that our Movement Behavior Model did not quantify the intensity of physical activity, which is something that was identified as important for cardiometabolic health and sleep quality [90,91]. Future studies should quantify heart rate while walking and performing other physical activity for exercise during the day (Figure 2). Further, it is possible that tracking work and leisure time metrics separately may facilitate more consistent movement throughout the day, thereby reducing prolonged sitting bouts. While further investigation is needed to ascertain the relative importance of the metrics presented here, these data and the Movement Behavior Model may serve as a general model for quantifying and further clarifying the metrics that are most important for optimizing musculoskeletal and cardiometabolic health among sedentary office workers.

Some considerations can be drawn on the basis of the obtained results. First, since time is limited in nature, spending more time on one activity (such as sitting, standing, or walking) will result in less time available for other activities. More time standing means less time sitting or walking. Increased time spent working reduces leisure time and the likelihood of physical activity during leisure time. This means that these measures are somewhat dependent on one another. Larger datasets could allow for a more robust statistical approach that accounts for data dependency, as well as their pattern of occurrence throughout the day. Perhaps, rather than the total amount of time spent on each activity, the timing, frequency, and duration of each occurrence are more important when studying their impact on health outcomes. For example, the association of posture transitions with MSD and cardiometabolic outcomes may differ if the analysis focuses on the pattern of posture transitions throughout the day instead of just the overall number of occurrences. The same could be true for patterns of sitting, standing, and walking. This should be explored in future studies. Some limitations of the study should be acknowledged. Data collection began at the end of 2019; therefore, given the COVID-19 pandemic, our sample size was limited. Larger populations are needed to fully understand the relationships investigated. Further, our sample of convenience from the university was a more active cohort than many sedentary workers, having a higher average standing time than cohorts in other studies [72]. More variance in the movement behaviors of participants may be needed in order to strengthen the observed correlations and to generalize the results presented. Moreover, as previously mentioned, the intensity of physical activity was not included in our analysis; its inclusion would likely improve the presented model and should be considered in future work (Figure 2). Furthermore, while our Movement Behavior Model features in-chair fidgeting, sway patterns, and mean pressure, these metrics were largely investigated in previous work [43]. Exploring all of these metrics together in a larger cohort may be needed to better understand the relative importance of each metric. Lastly, the cross-sectional design of this study did not allow for investigating causality between movement behavior measures, MSD, and cardiometabolic outcomes. Future research may include longitudinal studies to track changes in these parameters over time to better characterize the causal relationships that exist between them. This could contribute to the development of predictive models or activity scores that help sedentary workers take a comprehensive approach to increasing movement throughout the day.

## 5. Conclusions

This cross-sectional pilot study offers preliminary insights into the relationship between work and leisure time movement behavior measures, MSD, and cardiometabolic outcomes. The findings may be helpful for further study of these relationships and their application to interventions that prevent adverse health outcomes associated with sedentary behavior. Further investigation of a larger cohort may help to quantify the relative contribution of each measure and should consider the additional benefit from engagement in physical activities of different intensities. Ideally, the most important variables could be integrated into a user-friendly index, the impact of which could be investigated as an intervention.

## Figures and Tables

**Figure 1 ijerph-20-04668-f001:**
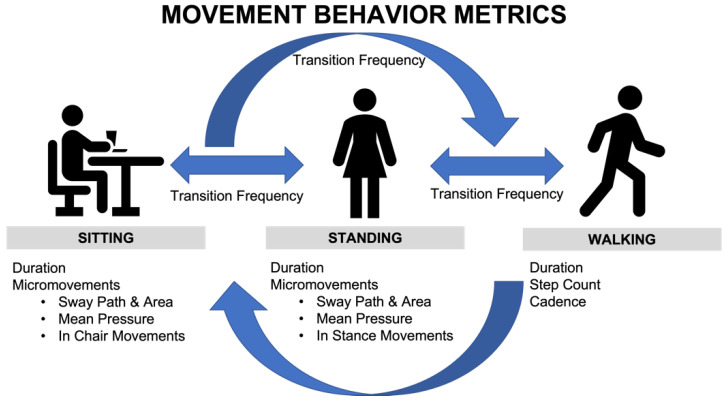
The comprehensive Movement Behavior Model for sedentary office workers. A transition is defined as a shift between two of sitting, standing, and walking. Each behavior was characterized by dedicated indicators.

**Figure 2 ijerph-20-04668-f002:**
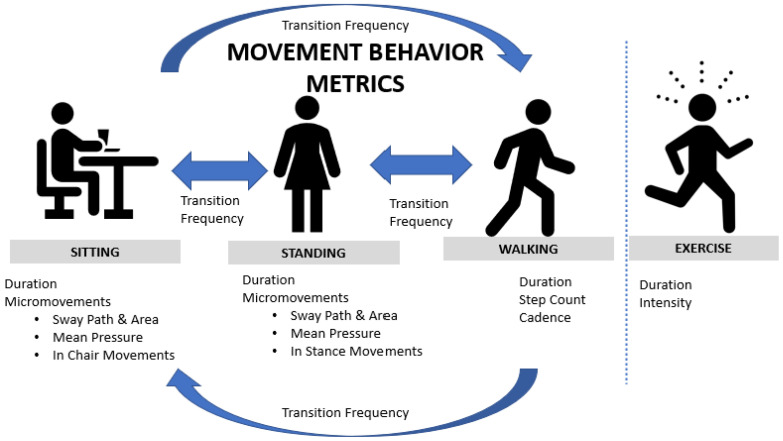
Revised comprehensive Movement Behavior Model that describes the different opportunities for movement throughout work and leisure time for sedentary office workers.

**Table 1 ijerph-20-04668-t001:** Demographics and cardiometabolic data of participants.

	All	MSD	NO-MSD
N	26	17	9
Gender *			
Male	9	8	1
Female	16	9	7
Other	1	-	1
Age (years) *	33.2 (9.4)	32.3 (7.9)	34.9 (12.1)
Children			
No	20	15	5
Yes	6	2	4
BMI	24.6 (3.9)	24.7 (3.5)	24.5 (4.9)
Hip:Waist ratio	1.2 (0.1)	1.2 (0.1)	1.2 (0.1)
Cardiometabolic data			
Resting heart rate (HR_rest_, rpm)	58.8 (9.7)	58.7 (9.2)	58.9 (11.2)
Average heart rate (HR_average_, rpm)	76.5 (10.4)	78.2 (9.6)	78.3 (11.7)
Average arterial pressure (MAP, mmHg)	87.4 (8.3)	86.7 (7.9)	88.8 (9.5)
Average pulse pressure (PP, mmHg)	42.4 (6.7)	41.4 (6.7)	44.1 (6.8)
Work hours	8.1 (0.5)	8.1 (0.5)	8.2 (0.4)

The symbol * identifies a significant difference between the two groups (*p* < 0.05).

**Table 2 ijerph-20-04668-t002:** Summary of musculoskeletal discomfort levels collected at baseline. Values are expressed as mean (standard deviation).

	All	MSD	NO-MSD
Head/neck/shoulders	1.5 (1.4)	2.2 (1.1)	-
Upper/lower back	1.9 (2.1)	2.9 (1.9)	-
Hips/knees/feet/ankle(s)	1.2 (1.5)	1.8 (1.5)	-
Elbow(s)/hands/wrists	1.2 (1.1)	1.7 (1.0)	0.1 (0.3)
Composite MSD	5.7 (5.0)	8.7 (3.5)	0.1 (0.3)

Data on MSD of the 4 different body regions are presented; these values were summed to provide a composite discomfort score.

**Table 3 ijerph-20-04668-t003:** Summary of the mean (SD) activity levels during work, leisure, and the entire day for the participants in the two groups.

	MSD (n = 17)			NO-MSD (n = 9)		
	Daily	Work	Leisure	Daily	Work	Leisure
Steps	9468.8 (2925.1)	5153.5 (2086.0)	4337.8 (1880.7)	10,055.8 (3556.8)	5810.3 (3146.0)	4245.4 (1594.2)
Transitions	52.5 (10.3)	25.4 (4.3)	27.1 (8.2)	49.6 (12.8)	22.7 (9.0) *	26.8 (6.6)
Hours standing	5.9 (1.7)	3.0 (1.1)	2.9 (1.1)	7.3 (2.4)	4.1 (1.6)	3.2 (1.0)
Hours walking	1.7 (0.5)	0.9 (0.3)	0.8 (0.3)	1.8 (0.6)	1.0 (0.5)	0.8 (0.2)
Hours sitting	10.0 (1.8)	6.0 (1.3)	4.0 (1.0)	9.2 (2.1)	5.1 (1.4)	4.1 (1.0)
Hours sleeping	8.5 (0.7)	-	-	8.1 (1.2)	-	-

* *p* < 0.05.

**Table 4 ijerph-20-04668-t004:** Spearman correlations between transitions, step count, and durations spent in various postures for those with composite musculoskeletal discomfort at baseline.

	Composite MSD		
	Daily	Work	Leisure
Steps	−0.26	0.10	−0.18
Posture transitions	0.38	0.46 *	0.20
Hours standing	−0.39 *	−0.27	−0.30
Hours walking	−0.31	0.09	−0.21
Hours sitting	0.20	0.29	0.13
Hours sleeping	0.21	-	-

Composite musculoskeletal discomfort was generated by summing the Numeric Pain Rating scale (0–10) across four regions. * *p* < 0.05.

**Table 5 ijerph-20-04668-t005:** Spearman correlations between the time spent standing, sitting, and walking during work and leisure time.

		Work				
		Steps	Transitions	Standing Time	Walking Time	Sitting Time
**Leisure**	Steps	0.04	−0.05	0.01	0.00	−0.07
	Transitions	−0.07	0.24	−0.26	−0.05	0.18
	Standing time	0.03	−0.29	0.21	−0.02	−0.31
	Walking time	−0.02	−0.19	0.01	−0.04	−0.05
	Sitting time	−0.42 *	−0.20	−0.64 **	−0.42 *	0.36
	Sleeping time	0.41 *	0.47 *	0.07	0.40 *	−0.05

* *p* < 0.05, ** *p* < 0.01.

**Table 6 ijerph-20-04668-t006:** Spearman correlations between activity levels and cardiometabolic data.

	Time	Hip:Waist Ratio	BMI	HR_Rest_	HR_Average_	MAP_Average_	PP_Average_
Steps	Daily	0.07	−0.09	−0.27	−0.23	−0.09	0.28
	Work	0.04	0.02	−0.17	−0.16	−0.11	−0.05
	Leisure	−0.02	−0.21	−0.34	−0.29	0.06	0.55
Transitions	Daily	−0.08	−0.10	−0.43 *	−0.58 **	−0.40	0.04
	Work	0.12	0.29	−0.28	−0.33	−0.43	0.18
	Leisure	−0.22	−0.39 *	−0.44 *	−0.52 **	−0.34	−0.04
Standing time	Daily	0.16	−0.05	−0.09	−0.19	0.22	−0.08
	Work	0.18	0.03	−0.14	−0.14	0.10	−0.33
	Leisure	−0.15	−0.28	0.06	−0.25	0.04	0.28
Walking time	Daily	0.12	−0.12	−0.21	−0.23	−0.01	0.33
	Work	0.08	−0.05	−0.17	−0.13	−0.21	−0.18
	Leisure	0.05	−0.25	−0.20	−0.20	0.25	0.58 *
Sitting time	Daily	−0.16	−0.07	0.09	0.16	−0.11	0.36
	Work	0.14	0.16	0.15	0.23	−0.12	0.53
	Leisure	−0.38	−0.19	−0.08	0.06	−0.29	−0.04
Sleeping time	Daily	−0.23	0.50 **	0.03	0.16	0.05	−0.32

* *p* < 0.05, ** *p* < 0.01.

## Data Availability

The data that support the findings of this study are available from the corresponding author upon reasonable request.

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
