# Peer review of "Movement Behavior and Health Outcomes among Sedentary Adults: A Cross-Sectional Study"

_ijerph, 2023, doi:10.3390/ijerph20054668_

Round 1
Reviewer 1 Report
This paper reports on a study entitled Exercise Behaviors and Health Outcomes in Sedentary Adults. I have reviewed the submitted manuscript and comment on the following points.
1. (Title)
This study is a compilation of cross-sectional studies. To avoid misleading the reader, it is necessary to clearly state "cross-sectional study" in the title. In addition, I recommend that you use words that are more appropriate to the content of the study rather than the broad term "exercise behavior”.
2. (Line.170)
A detailed description of the recruitment of participants is needed. How many people applied and how many met the criteria should be added. How many data did you need for this study? Since you are referring to past efforts, it probably make some estimate of the sample size. Also, you need to indicate to what population the data are representative (it would be difficult to generalize the results of this study).
3. (Line.180)
The data collection time is stated as 48 hours, but in order to analyze the results as data, please provide the rationale for setting the time to 48 hours, citing the literature, etc.
4.(Table 4, 5, 6)
Correlations are described for each indicator, but in order to discuss these correlations, it is necessary to describe the characteristics of each indicator. Correlation assumes that both indicators have the property that when one increases, the other also increases (or decreases) in one direction or the other. Furthermore, as mentioned in section 2, the significance of the analysis results should not be reflected in the discussion unless the sample size is considered.
5. (Line. 282)
As in 4., the significance of statistical significance should be clearly indicated.
6. (Line. 287)
If the NO-MSD group sat for 24 minutes per hour, does this mean that they sat for only 36 minutes per hour? If the total sitting time was 9 hours per day, how many hours of standing time would this mean?
7. (Line. 290)
You mentioned the possibility that participants with low back pain spent less time in the standing position. Please consider conducting the analysis again.
8. (Line. 297)
The recommendation to extend the walking time was not understood from the development of the text as to why it became a recommendation.
9.(Lines.318, 335, 343, 345, 348)
You describe the need for further investigation, but there seems to be a limit to what can be mentioned in this study. Why don't you consider reporting the results after further investigation (follow-up survey) is conducted?
10.(Line.330)
I understand the relationship with sleep duration, but are you dealing with sleep quality in this study? If you need to make this statement, you need to show the results of sleep quality and physical activity.
11. (Discussion)
As pointed out at the beginning of the text, the discussion and conclusion of this study seem to indicate a causal relationship, despite the fact that it is a cross-sectional study. In order to clarify this relationship, it is suggested that a follow-up study be conducted.
Reviewer 2 Report
First of all, I thank you for the opportunity to review this interesting contribution. This is an important issue to explore that i believ will merit publication. The aims of this study were to examine the movement behavior of sedentary office workers during both work and leisure time to explore its association with musculoskeletal discomfort (MSD) and cardiometabolic health indicators. There are some issues that need to be considered.
Introduction
- Provide recommendations on physical activity, World Health Organization, 2020
Participants
- Berkeley Committee for Protection of Human Subjects – can you add the consent number?
Procedure
- Were participants informed that they could withdraw without consequence at any time during the study?
- Line 183: anthropometric measurements – Where were the measurements taken? And what procedures were followed? Please describe your measurements in detail. Please indicate the protocol for taking height and weight data. Also include the brand of each instrument used.
Measures
- Regarding your variables analysed, you should add reference value in the methods section (BMI).
Discussion
- Lines 259-270: unnecessary repetition of information.
Conclusion
- Lines 378-386: unnecessary repetition of information from the results and discussion in the conclusions.
- Another paragraph about the implications for this study could be included towards the end.
Reviewer 3 Report
First of all, I want to note that it has been a pleasure review your manuscript. I think this is an interesting work on the movement behavior metrics and their association with musculoskeletal discomfort and cardiometabolic health indicators to inform interventions that optimize health among sedentary workers, although the sample is too small to draw conclusive results. It would be better to specify that these are preliminary studies.
After reading in depth the manuscript, I would like to make some comments:
- Line 32-33. First sentence needs a supporting reference.
- Line 37 the point after the reference.
- Line 51. Point after reference,
- Line 63-71, 95, 101, 103. Watch the spacing between words. Review the whole manuscript, please.
- The name of figure 1 should be more concise. It seems to be a continuation of the text. And the caption of figure 1 is repeating the definition already discussed in the text. If you are going to explain what sway patterns are, you should be more concise. It is too long a legend.
- Line 144: “Although prior research has investigated.....” what research are the authors referring to?
- The objectives of the work are being specified and it is not the time at the end of the introduction to talk about the limitations they may have had in carrying out the study: “Though the COVID167 19 pandemic truncated the scope and duration of this study, we present data collected to inform future work..”
- The sample size that should have been taken into account in view of the statistical power mentioned in the section on statistical analysis has not been discussed.
- Table 2. The name of the table should be more concise and the data presented in the title of the table should be put in the legend, at the end of the table.
- There should be some space between the end of the table or captions and the main text. It makes it clearer to differentiate between what is table and what is text.
- In table 4, why is Daily Work Leisure daily in bold?
- Explain in table 5, as in table 4, what an asterisk or two means.
- It should be synthesised: “The symbols * and ** indicate significant differences for comparisons at baseline…” (for example * (p <0.01)
- In Discussion section is better to speak in a more impersonal way.
- In Conclusion section: This sentence should be put in discussion not in conclusion: “Our findings align with other studies emphasizing the importance of "sitting less and moving more", though here we find that this recommendation should pertain to both
work and leisure time. (in which the references of the studies referred to by the authors should be given.
-Author Contributions: All authors significantly contributed to this work approve the final version of the manuscript.
Specify with the initials of each author which part he/she worked on.
Round 2
